# Chitosan and Anionic Solubility Enhancer Sulfobutylether-β-Cyclodextrin-Based Nanoparticles as Dexamethasone Ophthalmic Delivery System for Anti-Inflammatory Therapy

**DOI:** 10.3390/pharmaceutics16020277

**Published:** 2024-02-16

**Authors:** Giuseppe Francesco Racaniello, Gennaro Balenzano, Ilaria Arduino, Rosa Maria Iacobazzi, Antonio Lopalco, Angela Assunta Lopedota, Hakon Hrafn Sigurdsson, Nunzio Denora

**Affiliations:** 1Department of Pharmacy–Pharmaceutical Sciences, University of Bari “Aldo Moro”, 70125 Bari, Italy; giuseppe.racaniello@uniba.it (G.F.R.); gennaro.balenzano@uniba.it (G.B.); ilaria.arduino@uniba.it (I.A.); rosa.iacobazzi@uniba.it (R.M.I.); antonio.lopalco@uniba.it (A.L.); angelaassunta.lopedota@uniba.it (A.A.L.); 2Faculty of Pharmaceutical Sciences, University of Iceland, 107 Reykjavik, Iceland

**Keywords:** cataract surgery, mucoadhesive nanoparticles, dexamethasone, ocular delivery

## Abstract

Cataract surgery interventions are constantly increasing, particularly among adult and elderly patients. This type of surgery can lead to inflammatory states of the ocular anterior segment (AS), usually healed via postoperative treatment with dexamethasone (DEX)-containing eye drops. The application of eye drops is challenging due to the high number of daily administrations. In this study, mucoadhesive nanoparticles (NPs) were formulated to improve the residence time of DEX on the corneal mucosa, enhancing the drug’s solubility and bioavailability. The NPs were generated using an ionotropic gelation technique, exploiting the interaction between the cationic group of chitosan (CS) and the anionic group of sulfobutylether-β-cyclodextrin (SBE-β-CD). The formation of the inclusion complex and its stoichiometry were studied through phase solubility studies, Job’s plot method, and Bi-directional transport studies on MDCKII-MDR1. The obtained NPs showed good chemical and physical characteristics suitable for drug loading and subsequent testing on animal mucosa. The DEX-loaded CS/SBE-β-CD NPs exhibited a prolonged residence time on animal mucosa and demonstrated enhanced drug permeability through the corneal membrane, showing a sustained release profile. The developed NPs posed no irritation or toxicity concerns upon local administration, making them an optimal and innovative drug delivery system for inflammatory AS diseases treatment.

## 1. Introduction

Cataracts are one of the world’s major eye diseases, characterized by a gradual opacification of the eye lens, which can lead to complete blindness of the patient [1], and it affects between 11.8% and 18.8% of the world’s population [2]. Its incidence increases with age, ranging from 3.9% among 55–64-year-olds to 92.6% among those 80 years and older [3]. Many techniques have been used to treat the disease, including intraocular lens (IOL) placement [4], femtosecond laser-assisted surgery (FLACS) [5], and other pharmacological techniques such as the use of lanosterol to recover lens transparency [6]. To date, despite pharmacological techniques such as lanosterol and vitamin D or compounds possessing antioxidant and free-radical scavenging activity having shown promising potential in experimental studies, none of these drug candidates have translated into FDA-approved anti-cataract eye drops or remedies capable of preventing or treating cataracts in humans [7]. Therefore, surgery remains the main and most effective option for the treatment of this disease. However, surgical intervention can easily lead to an acute state of inflammation of the ocular anterior segment (AS) on postoperative days, which occurs with diffuse corneal edema and results in diffuse corneal endothelial damage [8]. Furthermore, when treating patients of different age groups, changes in the involved ocular tissues that may affect the incidence of the inflammatory state must also be considered. Compared to other ocular structures, the cornea does not show significant changes with normal ageing. However, clinically evident tissue changes are present that are asymptomatic and do not affect vision, making them difficult to detect. The aged cornea becomes more susceptible to infection due to a decreased ability to withstand a variety of physiological stresses [9]. In fact, the thickness of the cornea tends to flatten out with age, with a conspicuous loss of corneal endothelial cell density in an advanced age, thus leading to an increased risk of incurring inflammatory and infectious states following traumatic events such as cataract surgery [10]. Considering the extremely high number of such surgeries worldwide, perioperative organization and prophylaxis are important to improve the recovery time and quality of the patient’s response to surgery [11]. A reduction in AS inflammation resulting from surgical practice contributes to improved final visual outcomes in patients recovering from surgery [12].

Dexamethasone (DEX) is commonly used in the clinical treatment of patients who have undergone cataract surgery [13]. DEX is a fluorinated synthetic corticosteroid and is currently the most potent synthetic analog of cortisol [14]. In the treatment of ocular AS, DEX prevents and treats retinal vein occlusion and noninfectious uveitis, greatly reducing the postoperative inflammatory phenomenon [15,16].

Most commercially available medications for the treatment of inflammatory states in AS are provided as eye drops [17], which include DEX-loaded Drug Delivery Systems (DDS) [18]. Indeed, apart from instances of DEX being loaded into solid DDS like tablets or intravitreal implants [19], the most commonly used dosage forms currently available on the market are eye drops in which the drug is in suspension, either alone or in the presence of other active ingredients (e.g., levofloxacin, tobramycin, netilmicin). Formulations to date on the market have a 0.1% *w/v* of DEX, and the usual dose of DEX-based eye drops is one drop per eye repeated every 4 h. The administration of eye drops formulations is particularly challenging [20] due to the presence of static ocular barriers, such as anatomical (cornea, sclera, retina), haemato-aqueous, and haemato-retinal barriers. Moreover, the presence of dynamic barriers, such as the nasolacrimal duct, blinking reflex, and the low receiving volume of the conjunctival sac (20–50 µL), greatly reduce the bioavailability of drugs at the ocular AS [21]. In addition, adherence to therapy remains a critical issue, especially in the adult and elderly populations, who constitute the largest proportion of those exposed to this type of surgery [22]. The use of eye drops can represent an obstacle for these patients, given the difficulty in instilling the drops in multiple and repeated administrations at different times of the day. Thus, there is a need to formulate a safe, manageable, and effective product that can ensure the resolution of inflammation and optimal adherence to therapy [20].

Nanoparticles (NPs) have proven useful in overcoming static and dynamic ocular barriers by improving the pharmacokinetic properties and bioavailability of the drug and protecting it from physical, chemical, and biological degradation phenomena [23]. The advantages of nanosystems used as ophthalmic drug delivery systems also include increased permeability and bioavailability of the drug, increased retention of the drug on the surface of the eye, and improved interaction with the mucous membrane of the cornea [24]. In fact, smaller particles, such as nanoparticles, are well tolerated and can exhibit adhesive characteristics which are responsible for the prolonged contact time of a drug with the ocular tissue and its increased bioavailability [25]. In addition, the small size of NPs allows for the crossing of physical ocular barriers and facilitates drug release at the target site [26].

CS is a linear polysaccharide produced by the deacetylation of chitin and is commonly used in the formulation of CS-based NPs for medical purposes [27]. The polycationic nature of chitosan ensures strong ionic interactions with negatively charged components present on the surface of the biological mucosa, such as sialic acid [28], making CS a strongly mucoadhesive polymer [29].

However, DEX is known to be a substrate of P-glycoprotein (P-gp) [30] and induces P-gp overexpression in human retinal pigment epithelial cells after prolonged instillation [31]. P-gp is an efflux pump that hinders drug adsorption, and its overexpression is often present in inflamed tissues and has been reported to be associated with chemoresistance and poor prognosis in several ocular malignancies [32]. Thus, overexpression of P-gp in ocular tissues would result in the rapid clearance of DEX from the ocular surface if it were directly instilled as a suspension in an eye drop. Moreover, lipophilic molecules such as DEX (LogP = 1.83) often present a persistent challenge due to their low aqueous solubility which reduces their bioavailability and limits their therapeutic use [33].

In previous studies, it has been shown that cyclodextrins are poor substrates for P-gp [34], and, specifically, SBE-β-CD causes a direct inhibition of P-gp ATPase [35]. Moreover, by establishing inclusion complexes with hydrophobic drug molecules, the CDs are frequently used in DDS to improve the water solubility of poorly soluble drugs [36,37]. SBE-β-CD is a polyanionic derivative of β-CD characterized by the presence of a sodium sulfonate salt separated from the lipophilic cavity by a butyl ether spacer group [38]. The negative-charge repulsive forces of the terminal groups extend the cavity of this cyclodextrin, making it preferable for its stronger drug binding [39] and resulting in it being exploitable for interaction with polycationic chains in the NPs production process via ionotropic gelation. The inclusion complex formation has been used to enhance the stability of DEX in the ocular environment and permeability through the cornea owing to the higher solubility in the lacrimal fluids and also to the inhibition of P-gp exerted by SBE-β-CD [40]. Moreover, DEX/cyclodextrins inclusion complexes in association with CS have exhibited improved drug solubilization, enhanced mucoadhesive characteristics, and low toxicity in ophthalmic use [41].

The aim of this study was to develop novel nanotechnology-based eye drops for ophthalmic delivery of DEX, generating, for the first time, an SBE-β-CD/DEX inclusion complex-based formulation eligible as novel anti-inflammatory therapy post cataract surgery. In this work, a detailed investigation was carried out on the inclusion complex formation pathways between SBE-β-CD and DEX, with a focus on drug-CD interactions. A detailed characterization of the chemical–physical properties and mucoadhesive characteristics of NPs was carried out. In addition, the drug release and permeability were evaluated. These aspects were exploited to increase the residence time of the drug on the ocular mucosal surface, achieving a controlled release of the drug and decreasing the number of daily administrations of the eye drops when compared to commercially available formulations.

## 2. Materials and Methods

### 2.1. Materials

Low-molecular-weight chitosan (139.9 kDa, degree of deacetylation: 0.9), medium-molecular-weight chitosan (445.8 kDa, degree of deacetylation: 0.9), high-molecular-weight chitosan (749.4 kDa, degree of deacetylation: 0.9), and oligomer chitosan (13.49 kDa, degree of deacetylation: 0.8) were kindly donated from Primex EHF (Siglufjordur, Iceland). Sulfobutylether-β-Cyclodextrin sodium salt (M_w_ 2163, degree of substitution 6.5) (Captisol^®^) was purchased from CyDex Pharmaceuticals (Lenexa, KS, USA). Hydroxypropyl-β-cyclodextrin and dexamethasone were purchased from Farmalabor Srl (Canosa di Puglia, Italy). Calcium chloride dihydrate, polysorbate 80, potassium bromide, potassium chloride, potassium phosphate monobasic, sodium bicarbonate, sodium chloride, and sodium phosphate dibasic were purchased from Sigma-Aldrich–Merck Italy (Milano, Italy). MilliQ water used in the assays was obtained through a Milli-Q instrument by Millipore Sigma (Burlington, MA, USA). The human MDR1-transfected MDCKII cell line (MDCKII- MDR1) was obtained from Dr. Piet Borst (Netherlands Cancer Institute, Amsterdam, The Netherlands) [42].

### 2.2. Phase Solubility Studies and Determination of Inclusion Complex Constant

A phase solubility study was performed following the Higuchi–Connors method [43] to confirm the increased solubility of the drug in the presence of SBE-β-CD, using a previously established protocol [44,45]. Briefly, 2 mL samples containing aqueous solutions at various concentrations of SBE-β-CD ranging from 0% to 5% (*w/v*) were prepared, and an excess of DEX was added to each solution. The resulting suspensions were sonicated for 2 min at 37 °C and stored in an orbital shaker (MaxQ™ 6000 Incubated/Refrigerated Stackable Shakers) at 25 °C at a constant oscillation of 30 rpm for 72 h. After reaching equilibrium, the samples were centrifuged at 10,000 rpm for 15 min (Hettich MICRO22R Eppendorf centrifuge), and the supernatant was analyzed after filtration through 0.45 µm cellulose acetate (CA) membrane filters. The amount of solubilized DEX in each sample was obtained via absorption reading on a UV–visible Spectrometer (Lambda35, PerkinElmer instruments) at a wavelength λ = 245 nm after proper dilution in H_2_O/MeOH 50/50 *v/v* solution. A calibration curve (R^2^ = 0.99984) was created by dissolving DEX in H_2_O/MeOH 50/50 *v*/*v* solution in a concentration ranging from 5.09 × 10^−3^ mM to 1.2 mM. Each sample was analyzed in triplicate.

Equation (1) was used to estimate the 1:1 inclusion constant of DEX: SBE-β-CD (*K*_1:1_):(1)$$K_{1:1}=\frac{p}{S_{0}\left(1-p\right)}$$
where *S*_0_ is the intrinsic aqueous solubility of DEX, and *p* is the slope of the phase solubility diagram.

### 2.3. Job’s Plot

Job’s plot method was applied to examine and confirm the stoichiometry of the DEX/SBE-β-CD complex determined via a phase solubility study. The DEX/SBE-β-CD inclusion complex in an aqueous solution was analyzed using the continuous variation method to verify stoichiometry [46]. Two equimolar solutions (1.0 × 10^−4^ M) of DEX and SBE-β-CD were prepared in MeOH/H_2_O (50/50 *v/v*). The difference between absorbance intensity with and without SBE-β-CD (ΔA) at 245 nm was determined using the mixtures of DEX/ SBE-β-CD having different molar ratios (from 0 to 1) but keeping the total molar concentration of the species in solution constant. The resulting curves, called Job’s plots, yield a maximum, the position of which indicates the ligand/drug ratio of the complex in solution.

The ΔA x [DEX] was plotted against *r*, where *r* was defined by Equation (2) as follows:(2)$$r=\frac{\left[DEX\right]}{\left[DEX\right]+[SBE-\beta -CD]}$$

### 2.4. ^1^H-NMR Analysis

^1^H-NMR spectroscopy was used to assess the formation of the inclusion complex between DEX and SBE-β-CD. Experiments were performed on DEX solubilized in D_2_O/CD_3_OD 50:50 V:V and with the inclusion complex prepared as described in the manuscript with proper modification. Briefly, a D_2_O solution of SBE-β-CD 1% *w/v* with an excess of DEX was prepared and stirred; after 24 h, the sample was centrifuged, filtered, and then diluted with CD_3_OD to obtain a solution of D_2_O/CD_3_OD 50:50 V:V. The spectra were recorded using an Agilent VNMRS 500 MHz spectrometer, and the ^1^H-NMR spectra were analyzed using MestReNova version 6.0.2-5475; the residual solvent signal of deuterated water at 4.79 ppm was used as a reference for all the samples (Appendix A).

### 2.5. CS/SBE-β-CD NPs Preparation

The NPs were prepared using the ionotropic gelation technique (Figure 1) between polycationic chitosan (CS) and a polyanionic polymer, such as sulfobutylether-β-cyclodextrin (SBE-β-CD) [47,48]. For the NPs preparation, a previously described procedure was followed [37]. Briefly, a solution of 1% *w/v* SBE-β-CD was added dropwise into a glacial acetic acid solution (1% *v/v*) containing 0.5% *w/v* CS and 0.5% *v/v* Polysorbate 80, used as surfactant. Drops were added every 5 s under vigorous magnetic stirring at RT, and the resulting NPs suspension was left stirring for 30 min. Various volume ratios of the solutions were tested in the CD/CS ratio presented in Table 1. Furthermore, four different types of CS were tested for NPs preparation: low-molecular-weight CS (139.9 kDa), medium-molecular-weight CS (445.8 kDa), high-molecular-weight CS (749.4 kDa), and oligomer CS (13.49 kDa). Obtained NPs were isolated by centrifugation at 13,000 rpm for 30 min at 25 °C; then, the supernatant was removed, and the obtained precipitate was resuspended in distilled water.

### 2.6. DEX-Loaded NPs Preparation

NPs encapsulating DEX were prepared following the above-described methods, using as starting material an already formed inclusion complex of SBE-β-CD/DEX. Hence, SBE-β-CD/DEX inclusion complex was obtained by adding an excess amount of the drug to a 1% *w/v* SBE-β-CD solution and keeping it under constant stirring for 72 h; the solution was then filtered through 0.45 µm CA membrane syringe filters, obtaining a clear solution as a result. After preparation, DEX-loaded NPs were washed twice using water to remove potential acetic acid residues from the samples in 2 cycles of centrifugation/filtration (SL 16 Centrifuge Series–Thermo Fisher Scientific, Waltham, MA, US) at 4000 G for 20 min at RT using Amicon Ultra-4 centrifugal filter (MWCO 50 kDa).

### 2.7. DEX Quantification Using HPLC Method

HPLC analysis was performed using a Shimadzu NexEra equipped with the following modules: CBM-40 system controller, LC-40D solvent delivery module (pump), SIL-40C autosampler, DGU-405 degassing unit, CTO-40C column oven, and SPD-M40 photodiode array detector. A specific HPLC method was used for the DEX quantification using a Zorbax Eclipse plus C18 250 × 45 mm column, pore size of 5 µm, and a guard column with MeOH:H_2_O (0.6% Triethylamine, pH 2.6 obtained with H_3_PO_4_) in the ratio of 65:35 (*v/v*) as mobile phase, wit a flow rate of 1.2 mL/min. The DEX retention time was 5.7 min, and a 254 nm wavelength was adopted. Each sample was injected in duplicate using a volume of 20 µL, and the amount of DEX was assessed using a calibration curve (DEX concentration range: 5.09 × 10^−5^ mM–0.12 mM, R^2^ = 0.9999) (Appendix A).

### 2.8. NPs Characterization

#### 2.8.1. Particle Size, PDI, and ζ-Potential

The average hydrodynamic diameter (Z-average) and size distribution (polydispersity index, PDI) of the obtained NPs were measured using dynamic light scattering (DLS) with a Zetasizer Nano ZS (Malvern Instruments Ltd., Worcestershire, UK). The DLS study was conducted after suitable dilution in distilled water of the resulting NPs and DEX-loaded NPs suspensions. The ζ-potential of samples was measured in triplicate using a Zetasizer Nano ZS with measurements taken at 25 °C after appropriate dilution of the NPs and DEX-loaded NPs suspensions samples with distilled water.

#### 2.8.2. Differential Scanning Calorimetry

Thermal analysis was conducted on the individual raw materials, namely, SBE-β-CD, CS, and DEX, on their physical mixtures and the prepared freeze-dried NPs using a Mettler Toledo DSC 822e Stare 202 system (Mettler Toledo, Greifensee, Switzerland) equipped with a thermal analysis automatic program, STARe Mettler Toledo. Samples (5–10 mg) were placed in a flat-bottomed aluminum pan and heated from −20 to 300 °C at a constant heating rate of 5 °C/min under a N_2_ flow atmosphere. An empty pan was used as a reference.

#### 2.8.3. Fourier-Transform Infrared Spectroscopy (FT-IR)

FT-IR analysis was performed for the same samples as mentioned above using a PerkinElmer 1600 FT-IR spectrometer (Waltham, MA, USA) on tablets obtained by dispersing 2% (*w*/*w*) of each sample in KBr for spectroscopy. The scan range investigated was 400–4000 cm^−1^, with a resolution of 1 cm^−1^.

#### 2.8.4. Encapsulation Efficiency of DEX in NPs

The encapsulation efficiency (EE) of DEX in the DDS was determined using a direct method. Briefly, the drug-loaded NPs were separated from the aqueous medium using the centrifugation–filtration technique at 4000 G for 20 min at RT using an Amicon Ultra-4 centrifugal filter (MWCO 50 kDa). Then, purified DEX-loaded NPs were dissolved in diluted HCl (0.5 M), and DEX released from broken NPs was quantified through the HPLC method. The *EE* (%) value was obtained using Equation (3) as follows:(3)$$EE\left(\%\right)=\frac{drug\;loaded\;in\;NPs}{total\;amount\;of\;drug}\times 100$$
where “*drug loaded in NPs*” represents the amount of active ingredient present in the NPs and quantified via HPLC measurements, while “*total amount of drug*” represents the amount of drug used during the NPs preparation stage and which is theoretically present in the formulation.

#### 2.8.5. Transmission Electron Microscopy (TEM)

TEM was employed to confirm the NPs’ size, shape, and morphology using a standard protocol [49]. In brief, a drop of the NPs formulation was carefully placed onto a lacey carbon-coated copper TEM grid (300 mesh). Subsequently, the grid was treated with 1% osmium tetroxide for 1 min before being rinsed with ultrapure water and allowed to dry. A JEOL Jem1011 microscope (Tokyo, Japan) operating at an accelerating voltage of 100 kV was used to capture the resulting image.

### 2.9. Preparation of Bovine Eye Cornea

Bovine eyes from an adult bovine were obtained fresh from a local slaughterhouse (S.P. Palo del Colle–Bitonto, Bari, Italy) and placed in an ice bath, and the tests were performed within 3 h from the excision. The eyes underwent washing processes with simulated tear fluid (SLF). The composition of used SLF was sodium chloride (0.670 g), sodium bicarbonate (0.200 g), and calcium chloride dihydrate (0.008 g) in 100 mL of purified water [50]. The pH of obtained SLF was adjusted with diluted HCl to a set value of 7.4, while the osmolarity of the solution was 277.62 mOsm/L (normal tear osmolarity ≤ 308 mOsm/L). After external cleaning, the excised bovine eyes were stored in the freshly prepared SLF solution (pH 7.4) to preserve their integrity. The bovine eyes were then cut to expose the cornea. A final 1 cm × 1 cm piece was then collected from the eye and washed before use.

### 2.10. Ex Vivo Mucoadhesive Studies

The mucoadhesion wash-off test was performed to measure the NPs’ mucoadhesive properties following a slightly modified protocol [51]. For the assay, 500 μL of the NPs sample (containing 435 μg of DEX) was applied to the cornea of the bovine eyes placed on a 45° inclined plane and washed with 120 mL of STF pH 7.4 at a flow rate of 1 mL/min using a Legato Syringe pump (KD Scientific, Holliston, MA, USA). The apparatus is illustrated in Figure 2. A sample containing the same amount of DEX in 2% *w/v* HP-β-CD was used as standard control. At 2, 5, 10, 15, 20, 30, 40, 50, 60, 80, 100, and 120 min, 300 μL of the washing solution was withdrawn and replaced with an equal volume of fresh STF solution. Each collected sample was treated with diluted HCl (0.5 M) to dissolve the NPs detached during washing and allow for the reading of released DEX through the HPLC method. The amount of DEX attached to the bovine cornea was plotted against the wash-off time.

### 2.11. In Vitro and Ex Vivo Permeation Studies

In vitro and ex vivo permeation studies were conducted to verify the membrane-crossing behavior of DEX. For this purpose, two different barriers between donor and receptor chambers in Franz’s cell with a diffusion area of 0.6585 cm^2^ were used, the former consisting of a Spectra/Por 3 standard regenerated cellulose membrane (MWCO 3.5 kDa) [52] and the latter of a corneal tissue taken from a bovine eye. The donor chamber was alternatively filled with 500 μL of DEX-loaded NPs (containing 435 µg of DEX) or 500 μL of a solution containing the same amount of DEX in 2% *w/v* HP-β-CD, as standard control. An acceptor medium consisting of 9.5 mL of PBS (pH 7.4) with 2% *w/v* HP-β-CD was placed in the receptor chamber to ensure proper solubility of DEX. The systems were placed in a 37 °C thermostatic bath to reproduce in vivo conditions, and 300 μL of the acceptor medium was withdrawn at regular time intervals. Replacements with a fresh medium were performed to maintain sink conditions. The obtained samples were analyzed using the HPLC method to measure the amount of DEX permeated over time. Five conventional release models, including zero-order, first-order, Higuchi, Hixson–Crowell, and Korsmeyer–Peppas, were fitted to the experimental data, and regression analysis was performed in each case (equations used are shown in Table 2) [53,54]. All data were processed using Microsoft Excel version 2211, and the best-fitting models were considered [55]. The R^2^ and the n values were used as indicators of the model’s suitability for the given dataset.

The steady state flux (*J*) was calculated as the slope of linear plots of the amount of drug in the receptor chamber (*Q*) versus time and the apparent permeation coefficient (*P_app_*) determined from Equation (4):(4)$$J=\frac{dQ}{Axdt}=P_{app}\times C_{d}$$
where *A* is the surface area of the mounted membrane (0.6358 cm^2^), and *C_d_* is the concentration of dissolved drug in the donor chamber.

### 2.12. Bi-Directional Transport Studies on MDCKII-MDR1

To assess the ability of SBE-β-CD’s complexation on permeability improvement by inhibition of P-gp, apical to basolateral (*P_app_, AP*) and basolateral to apical (*P_app_, BL*) permeability studies of the obtained samples were performed using Madin-Darby Canine Kidney (MDCK) cells retrovirally transfected with human MDR1 cDNA (MDCKII-MDR1), as previously reported [56]. Cells were cultured in a DMEM medium (EuroClone, Pero (MI), Italy) and seeded at a density of 100,000 cells/cm^2^ on 12-well polyester Transwell inserts (pore size 0.4 μm, diameter 12 mm, apical volume 0.5 mL, basolateral volume 1.5 mL). At first, the MDCKII-MDR1 cell barrier function was verified through trans-epithelial electrical resistance (TEER) using an EVOM apparatus and through the measurement of the flux of the paracellular standard fluorescein isothiocyanate-dextran (FD4, Sigma-Aldrich Merck Italy, Milano, Italy) (200 μg/mL) and the transcellular standard diazepam (75 mM) [56]. Cells were equilibrated in a transport medium in both the apical and basolateral chambers for 30 min at 37 °C. The composition of the transport medium was as follows: 0.4 mM K_2_HPO_4_, 25 mM NaHCO_3_, 3 mM KCl, 122 mM NaCl, 10 mM glucose with a final pH of 7.4, and the osmolarity was 300 mOsm as determined by a freezing point-based osmometer. At time 0, the culture medium was aspirated from both the apical (AP) and basolateral (BL) chambers of each insert, and the cell monolayers were washed three times (10 min per wash) with Dulbecco’s phosphate-buffered saline (DPBS) pH = 7.4. Finally, a solution of compounds diluted in the transport medium was added to the apical or basolateral chamber. For AP-to-BL or BL-to-AP flux studies, the drug solution was added to the AP or BL chamber, respectively. The compounds were first dissolved in DMSO and then diluted with the test medium to an appropriate concentration. The tested solutions were then added to the donor side (500 µL for the AP chamber and 1.5 mL for the BL chamber), and a fresh assay medium was added to the receiver compartment. The amount of DMSO in the samples never exceeded 1% (*v/v*). Transport experiments were performed under cell culture conditions (37 °C, 5% CO_2_, 95% humidity). After an incubation period of 120 min, samples were taken from the apical and basolateral sides of the monolayer and stored until further analysis. A quantitative analysis of DEX in the tested samples was performed using the HPLC method reported above. Each compound was tested in triplicate, and experiments were repeated three times. The apparent permeability, in units of cm/sec, was calculated using the following Equation (5):(5)$$P_{app}=\left(\frac{V_{a}}{Area\times time}\right)\times \left(\frac{\left[drug\right]acceptor}{\left[drug\right]initial}\right)$$
where “*V_ɑ_*” is the volume in the acceptor well, “*Area*” is the surface area of the membrane, “time” is the total transport time, “*[drug] acceptor*” is the concentration of the drug measured using HPLC, and “*[drug]initial*” is the initial drug concentration in the AP or BL chamber.

Efflux ratio (*ER*) was calculated using the following Equation (6):(6)$$ER=\frac{P_{app},BL-AP}{P_{app},AP-BL}$$
where *Papp*, *BL* − *AP* is the apparent permeability of basal-to-apical transport, and *Papp*, *AP* − *BL* is the apparent permeability of apical-to-basal transport. An efflux ratio greater than 2 indicates that a test compound is likely to be a substrate for P-gp transport.

### 2.13. Ocular Irritation Test

To confirm the non-irritancy and toxicity of the formulation, a slightly modified Bovine Corneal Opacity and Permeability (BCOP) assay was performed [57]. For the test, 750 µL of 0.1 N NaOH solution as a positive control (irritant), 750 µL of 0.9% NaCl as a negative control (non-irritant), and 750 µL of DEX-loaded NPs sample, containing 650 µg of DEX, were applied to the cornea of three bovine eyes, held horizontally and stored in a 0.9% NaCl solution in a 37 °C thermostatic bath. At 0, 60, 120, 180, and 240 min, photos of the eyes were taken, and opacity of the corneas was visually evaluated. An assessment of the thickness of the cornea was carried out using a Hitech Diamond digital gauge. An irritation score, based on the opacity caused on the cornea upon application and on the increment in width of the cornea, was calculated and assigned to each substance [58].

## 3. Results and Discussion

### 3.1. Characterization of DEX/SBE-β-CD Inclusion Complex

The DEX/SBE-β-CD inclusion complex was necessary to increase the residence time of the drug on the corneal membrane and to enhance the aqueous solubility and bioavailability of the drug. To the best of our knowledge, the formation of the inclusion complex and its stoichiometry were extensively studied for the very first time in this work through the phase solubility studies, Job’s plot method, and ^1^H-NMR analysis. As reported in Figure 3A, the DEX/SBE-β-CD complex shows a maximum value at r = 0.5, with a symmetrical shape proving the presence of 1:1 stoichiometry, within the range of evaluated concentrations. The aqueous solubility (S_0_) of DEX at 25 ± 0.2 °C was recorded in ultrapure water and evaluated through HPLC analysis, quantifying the concentration of a saturated solution of the drug. In accordance with the data already reported, the aqueous solubility of DEX is equal to about 0.1 mg/mL [59]. The phase solubility study is a widely accepted method to estimate the effect of cyclodextrin complexation on the solubility of a drug, and the apparent constant of complexation (*K*_1:1_) can be calculated using the slope and the intrinsic solubility (*S*_0_ = 0.11 mg/mL) of the drug obtained from the plotted data (Figure 3B). The drug solubility increased in the presence of SBE-β-CD with a maximum of 70-fold improvement in the observed range (0.003–0.035 M) and a maximum of 10-fold improvement in the used SBE-β-CD concentration (0.014 M) for NPs production. According to Higuchi and Connors’ equation, an A_L_ type solubility diagram was observed, characterized by a line with a slope lower than the unit and a complexation constant (*K*_1:1_) for DEX/SBE-β-CD equal to 1242.28 M^−1^ [60]. The NMR data presented in Appendix A confirm the inclusion complex’s formation. In fact, from obtained information, we can suggest that the DEX molecule can be totally or partially enclosed in the CD cavity. There could be an interaction between the aromatic ring of the DEX inside the CD because the H-C, H-D, and H-E protons results were deshielded, confirming the interactions between the molecules and the formation of the inclusion complex. The observed behavior is similar to what is already documented in the literature regarding the complexation between DEX and β-CD [61]. In our case, the Δ(δDEX/SBE-β-CD and δDEX are lower due to a lower complexation capacity of SBE-β-CD (*K*_1:1_ DEX/SBE-β-CD = 1242.28 M^−1^) compared to that of β-CD (*K*_1:1_ DEX/β-CD > 2000 M^−1^) [62].

### 3.2. Nanoparticles Preparation and Characterization

Optimal conditions for NPs preparation were set using 1 mL of 1% *w/v* SBE-β-CD solution and 1.4 mL of 0.5% *w/v* CS in 1% *v/v* glacial acetic acid solution, under constant magnetic stirring at 700 rpm at 25 °C. The alternative protocol set using 2 mL of 1% *w/v* SBE-β-CD and 5 mL of 0.5% *w/v* CS in 1% *v/v* glacial acetic acid solution was not successful in forming stable monodispersed nanoparticles. Therefore, after an initial study involving the characterization by DLS of multiple preparations obtained within a ratio range encompassing the two proposed ratios, only the first protocol was considered a suitable procedure. In fact, NPs produced with a different ratio were discarded as they were found to be unstable and prone to aggregation, showing a size and PDI value too high to be considered exploitable. Polysorbate 80, added to the CS solution before NPs preparation, prevented surface interaction and, consequently, aggregation phenomena. As shown in Table 3, the low-molecular-weight CS generated NPs with a diameter of 210.3 nm and a PDI of 0.161, suggesting a favorable mono dispersion of the system. In contrast, using medium- and high-molecular-weight CS resulted in excessively large particles and aggregates (size > 342 nm, PDI > 0.327), while no particle formation was observed with the use of Oligomer CS in the adopted conditions. In addition, from the evaluation of the ζ-Potential, it was possible to see that a strongly positive value was associated with NPs prepared from low-molecular-weight CS (ζ = +29.4), indicating good stability of the nanoparticle system, compared to the other two nanoformulations [63]. These results highlight the influence of the proper molecular weight selection. In fact, the use of a low-molecular-weight CS ensures a major exposure of the cationic groups involved in the interaction with the mucosa and avoids the possible phenomena of aggregation, maintaining optimal diameter and polydispersion values [64]. As shown in Table 3, the DEX-loaded NPs obtained from Low M_w_ CS showed comparable results to those of unloaded Low M_w_ CS, showing an optimal diameter value useful for permeation into biological membranes, and low polydispersion and high ζ-potential values, indicating good stability and a potential mucoadhesive nature of the system. Furthermore, the evaluation of EE highlighted the efficiency of the encapsulation protocol used, which resulted in a loaded drug rate of more than 87% (Table 3).

Therefore, all the characterization studies were performed on NPs based on low-molecular-weight CS.

The samples obtained from the low-molecular-weight CS showed the best dimensional, chemical–physical, and stability characteristics. The dimensional characteristics of all obtained NPs align with the typically used nanoparticle sizes in the ocular field. However, it is worth considering that smaller-sized NPs provide a larger specific surface area for interaction with the mucin layer, significantly enhancing the mucoadhesive properties of the formulation [65]. Moreover, smaller-sized NPs facilitate mucopermeation through mucosal barriers, ensuring drug release into the underlying tissues [66]. The low molecular weight CS NPs particularly exhibited a low PDI value, ensuring the homogeneous dispersion of NPs. For all these reasons, it was decided to continue with further testing only on this sample.

### 3.3. Characterization of DEX-Loaded NPs

The DSC analysis was carried out to evaluate the thermic behavior of the DEX/SBE-β-CD inclusion complex encapsulated in the medicated NPs. The FT-IR study was performed to confirm the presence and structural integrity of CD within the NPs and to reveal the interactions between the SBE-β-CD and DEX. TEM analysis provided more information about the size, shape, and morphology of DEX-loaded NPs.

As shown in Figure 4, DSC analysis of DEX showed a characteristic endothermic peak at 265 °C, corresponding to the melting point of the drug. The absence of this peak in the thermogram of DEX-loaded NPs suggested that the drug was encapsulated in the toroidal structure of SBE-β-CD. As a control, the same analysis was repeated on all the components of NPs (Appendix A) and on the physical mixture of all components of the final formulation (Figure 4), and the obtained diagram showed the presence of the characteristic endothermic peak of DEX.

The FT-IR spectrum of DEX (Figure 5) showed the characteristic absorption bands of the drug at 3390 and 1268 cm^−1^ caused by the stretching vibration of O-H and C-F bonds, respectively. The stretching vibration in 1706, 1662, and 1621 cm^−1^ was due to the -C=O and double-bond framework conjugated to -C=O bonds [67]. In pure CS, two characteristic absorption bands at 1634 and 1538 cm^−1^ were detected and attributed to the amide -C=O group and N-H (amine) group vibration, respectively. Also, a wide overlapping absorption band was observed around 3250 cm^−1^ due to the stretching vibration of O-H bonded to N-H in the CS spectra [67] (Appendix A). The IR spectrum of SBE-β-CD showed strong absorption bands at 3423.46 cm^−1^ (O-H stretching), 2937.51 cm^−1^ (C-H stretching), 1645.60 cm^−1^ (δ-HOH bending of water molecules attached to CD), 1161.37 cm^−1^ (C-H vibrations), and 1043.24 cm^−1^ (C-O stretching) [68] (Appendix A).

The DEX characteristic peaks were not observed, however, in the NPs spectrum, which was more comparable with the spectra of CS and SBE-β-CD, whose characteristic peaks mostly could also be found in the physical mixture spectrum. These results represent further evidence of the presence of the drug encapsulated in NPs as a DEX/SBE-β-CD inclusion complex.

The TEM image of DEX-loaded NPs (Figure 6) displayed a general distribution of NPs, characterized by a perfect spherical structure. Furthermore, the NPs in the image show a size range of diameters from 181 to 233 nm, which are comparable to the size values obtained from the DLS analysis.

### 3.4. Ex Vivo Mucoadhesive Studies

Ex vivo mucoadhesive studies were performed on water-dispersed NPs samples placed on the bovine eyes using the wash-off method. The evaluation of the mucoadhesion of the sample was based on the amount of DEX still present on the bovine ocular mucosa after constant washing with STF (pH 7.4) for a total time of 120 min.

Figure 7 shows that the DEX-loaded NPs exhibit an initial rapid decrease in the amount of drug present on the mucosa (first 15 min), followed by a retention of the drug over time up to 120 min, differing from the control response showing the complete absence of drug present on the mucosa after 10 min. In the first phase, a rapid decrease in the drug content on the mucosa was obtained, probably due to the removal of the drug found in the outer surface of the NPs during the washing off steps of the system. In the second phase, after about 30 min, a smaller decrease in the drug content over time is observed, probably due to the controlled release of DEX from the NPs systems adhering to the mucosa. This behavior is due to the interaction of the cationic layer consisting of CS covering NPs with the mucosal surface, which exhibits an anionic charge due to the presence of sialic acid [69]. The establishment of these bonds increases the mucoadhesive properties of the formulation, resulting in an enhancement of the resistance of the drug to external clearance processes and its residence time on the site of action, and potentially in a reduced number of daily instillations to the eye. In order to assess the quantity of drug released during the residence time of the formulation on the corneal surface and evaluate the potential of the obtained nanoformulation to reduce the number of ocular instillations, we subsequently conducted in vitro and ex vivo permeation tests.

### 3.5. In Vitro and Ex Vivo Permeation Studies

Drug permeation from NPs was evaluated in Franz cells in the presence of a control, using two types of membranes: the standard Spectra/Por 3 regenerated cellulose membrane (MWCO 3.5 kDa) and a corneal tissue adequately obtained from bovine eye [70]. As shown in Figure 8, NPs show a slower permeation, with a slightly longer retention time in the cornea, due to the different compositions and greater thickness of the crossed membrane. Within the first hour, NPs experienced a burst release effect, with 43.29% of the drug permeated through the cellulose membrane and 9.28% of the drug permeated through the cornea. The initial burst effect could be due to the matrix nature of the NPs produced, which, thus, exposes part of the DEX on the outer surface, undergoing more rapid solubilization of the drug [71]. This result also suggests that the CS present in NPs matrix underwent an apparent hydrolysis in solution, providing a relatively quick release of DEX in the first phase [72]. In the following hours, the NPs showed a lower percentage of permeated DEX in 24h (75.68%) than the control (79.21%) when crossing the cellulose membrane (Figure 8A). Instead, the NPs showed a higher DEX permeate value (17.53%) than the control (10.93%) when crossing the corneal membrane in the 6h evaluation (Figure 8B). This behavior may be attributed to the presence of a highly hydrophilic film, such as the tear film, above the corneal layer, which can act as a barrier in the case of a DEX release from the control, while it would represent an easily surmountable barrier in the case of the DEX-loaded NPs sample [73]. In fact, the enhanced retention of nanoparticles on the ocular surface due to nanoparticle size may be responsible for the increased penetration from the nanoparticles as compared to the control (marketed formulation) [23,74]. Furthermore, the presence of CS in the produced NPs may have contributed to enhancing the mucopenetration of the DEX-loaded NPs by opening mucin meshwork, facilitating the release of DEX into the underlying lipophilic layer [66]. As a result of this behavior, DEX-loaded NPs could be strongly recommended in the post-operative treatment of cataracts and, specifically, in the treatment of inflammatory states in older subjects. Indeed, these patients are characterized by a compromised corneal membrane with a thinned mucosal layer. This characteristic, which usually leads to a higher exposure of these patients to infection and inflammation, can be exploited by DEX-loaded NPs in order to achieve a higher mucopermeation of the formulation. This could be converted into a greater available amount of DEX in the underlying corneal layers and a more effective treatment of the inflammatory state.

Different kinetic models were applied to study the mechanism of the release process evaluating the R^2^ and *n* values obtained for each model. A bilinear kinetic was observed for both release studies. The release against membrane shows a Korsmeyer–Peppas release kinetic for the first 30 min, followed by a zero-order kinetic until 24 h. Similarly, the release against the corneal layer shows a Korsmeyer–Peppas-type kinetic for the first 60 min, followed by zero-order kinetic until 6 h (shown in Table 4). Evaluating the obtained values of n related to the Korsmeyer–Peppas kinetics (*n* > 1), the model obtained is of the Super Case II type, characterized by an initial swelling phase of the outer layer of the gel and a constant increase in tension and subsequent breaking of the polymer chains during the release phase, which can lead to a slowdown in the drug release of the samples [75]. Subsequently, zero-order kinetics indicate a constant release of the drug only as a function of time, independent of the concentration of the loaded actives [54].

The fluxes of permeated DEX per hour are presented in Table 5. The mucoadhesive NPs showed the highest *P_app_* values, with an enhancement ratio calculated as the ratio between the *P_app_* of the formulation and the *P_app_* of the standard control, being 2.04 for the corneal mucosa. Considering these results, the NPs obtained, due to their higher permeation properties, would be more suitable for the treatment of ocular diseases through the transcorneal administration.

### 3.6. Bi-Directional Transport Studies on MDCKII-MDR1

DEX is known to be a P-gp substrate that is efficiently effluxed, thus affecting corneal permeation and the systemic bioavailability of DEX [31]. SBE-β-CD modulate P-gp mediated efflux by inhibiting P-gp ATPase activity, which, in turn, improves permeability and, therefore, bioavailability [33,56]. The bi-directional transport studies were conducted to assess the possible interaction between SBE-β-CD and P-gp on biological membranes, leading to increased DEX transport across the corneal membrane by reducing its efflux. MDCKII-MDR1 cells were selected as model cells for the studies because they form tight monolayers and express P-gp, which is specifically involved in the efflux transport of drugs from biological membranes. Therefore, transport studies were conducted in both the AP-to-BL and BL-to-AP directions, and the results are shown in Table 6.

As seen from the obtained results, the permeation of free DEX in the P_app_ BL-AP direction across MDCKII-MDR1 cells was higher (1.87 × 10^−7^ cm/sec) than that in the AP-BL direction (0.66 × 10^−7^ cm/sec), with an ER value of 2.84, meaning that DEX was a substrate for P-gp (Table 6). This transcellular transport in the AP-BL direction was increased in the presence of the P-gp inhibitor SBE-β-CD. In fact, the DEX-loaded CS/SBE-β-CD NPs sample demonstrated a higher P_app_ (~1.4-fold) for the AP-BL transport compared to the free DEX sample. The efflux ratio (ER) value in the presence of SBE-β-CD was lower than 2 (0.92) and then, that in its absence, suggesting that the presence of SBE-β-CD can improve the bioavailability of the drug by interacting with P-gp and avoiding the elimination of DEX. The values obtained for Diazepam and FD4 used as transcellular and paracellular route markers were in agreement with those previously reported in the literature.

### 3.7. Ocular Irritation Test

The potential irritant reaction of the formulation was visually evaluated, applying a slightly modified version of the BCOP test [37]. An irritation score value was assigned to each substance tested on the cornea of the bovine eyes. The changes in opacity caused by the positive control application, after 4 h, were intense and spread, affecting the entire cornea, and a relative irritation score of 4 was assigned (Figure 9c,c′). In addition, the change in width (2.35 mm) showed an increase of more than 50% of the normal value (1.50 mm), thus exceeding the threshold (36%) for a substance considered highly toxic for ocular use [76]. On the other hand, as shown in Figure 9, the NPs formulation, as well as the negative control, caused no change in corneal opacity within 4 h after application, and no significant variation in corneal width was recorded. As a combination of the endpoints and assigned values, the positive control was designated as severely irritating, while the negative control of NaCl 0.9% and the NPs had no detectable consequences on the eyes with an assigned score of 0. Therefore, the NPs were classified as non-irritant and safe for topical use.

## 4. Conclusions

In this study, DEX-loaded NPs based on the ionotropic gelation of the cationic mucoadhesive chitosan and the anionic solubility enhancer SBE-β-CD were successfully developed. In fact, the poor water solubility of DEX was significantly improved through complexation with SBE-β-CD, as evidenced by the in-depth characterization of the obtained inclusion complex, throughout phase solubility studies and the ^1^H-NMR analysis. Subsequent evaluations of the physicochemical and solid-state properties confirmed that the drug was complexed and encapsulated with a high efficiency (>87%) within the NPs. Chemical and physical characteristics of the resulting DEX-loaded CS/SBE-β-CD NPs were assessed, and an ex vivo study on animal mucosa was carried out to verify the mucoadhesive properties of the final formulation. Indeed, mucoadhesive properties of the system showed that the CS coating allowed for the NPs to adhere to the conjunctival mucosa for up to 2 h after application, showing a substantial improvement compared to the commercial formulations. Moreover, the presence of CS also improved the mucopermeation of the formulation owing to its interactions with the mucin layer. In order to assess the potential utility of these new DEX NPs in the treatment of inflammatory diseases of AS, the ex vivo characterization of the animal corneal tissue was also carried out to investigate the drug permeability within the corneal layers. The system exhibited permeation through the corneal membrane demonstrating double the drug permeation compared to the standard control in the 6 h assessment period. Increased patient compliance could be achieved due to the longer residence time of the drug on the corneal surface, which leads to reduced drug loss, increased duration of action, and, thus, fewer daily administrations needed. Finally, an ocular irritability test was performed on the animal tissue through the BCOP assay to exclude any potential toxicities of the obtained DEX-loaded CS/SBE-β-CD NPs. An irritation test confirmed that the formulation was non-irritant and non-toxic upon application to the cornea, and, therefore, that the DEX-loaded CS/SBE-β-CD NPs was safe for the topical ophthalmic use of ocular inflammatory disease treatments. The conducted studies highlight that DEX-loaded CS/SBE-β-CD NPs could emerge as a novel and effective DDS for the postoperative management of AS inflammatory states derived from cataract surgery, proving to be preferable for the sustained treatment of the disease, potentially reducing the number of administrations. The formulation may also be strongly recommended in the treatment of AS inflammatory states in older subjects, as it may benefit the altered condition of the corneal membrane in these patients.

## Figures and Tables

**Figure 1 pharmaceutics-16-00277-f001:**
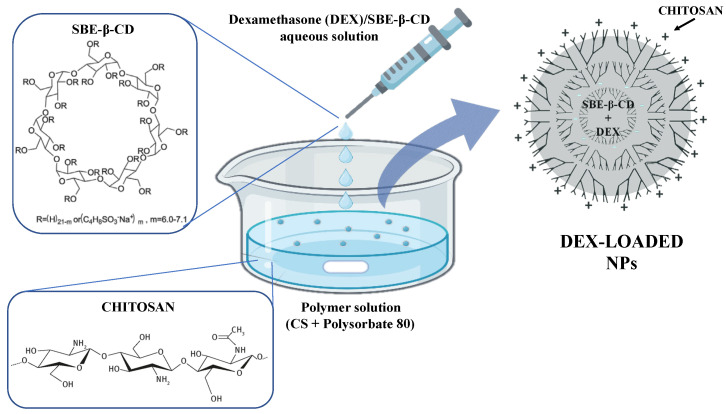
Schematic representation of DEX-loaded CS/SBE-β-CD NPs formation via ionotropic gelation technique.

**Figure 2 pharmaceutics-16-00277-f002:**
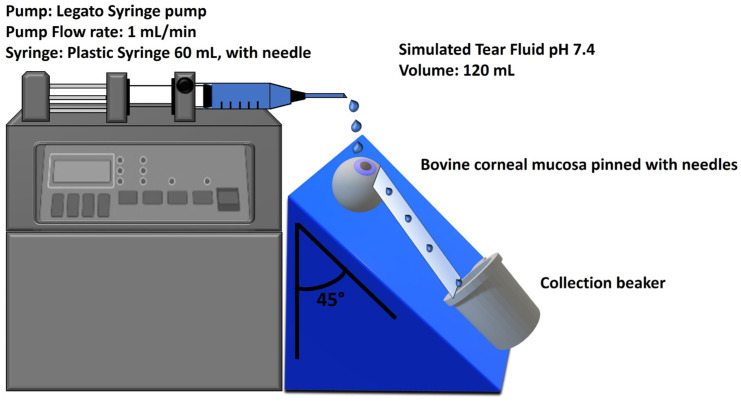
Bovine eye on a 45° inclined plane for mucoadhesion wash-off test.

**Figure 3 pharmaceutics-16-00277-f003:**
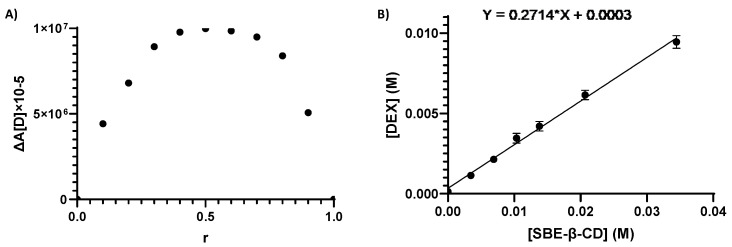
Job’s plot diagram for DEX/SBE-β-CD inclusion complexes’ stoichiometry determination (**A**); AL-type solubility curve, according to the Higuchi–Connors equation (**B**).

**Figure 4 pharmaceutics-16-00277-f004:**
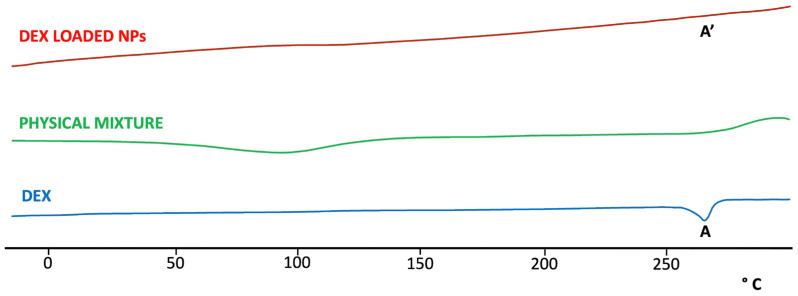
DSC thermo-analysis of DEX, DEX-loaded NPs, and physical mixture of all components in the NPs formulation. The characteristic endothermic peak present in the DEX thermogram (A) is entirely absent in the thermogram of the DEX-loaded NPs (A’), indicating the interaction of the drug with CD.

**Figure 5 pharmaceutics-16-00277-f005:**
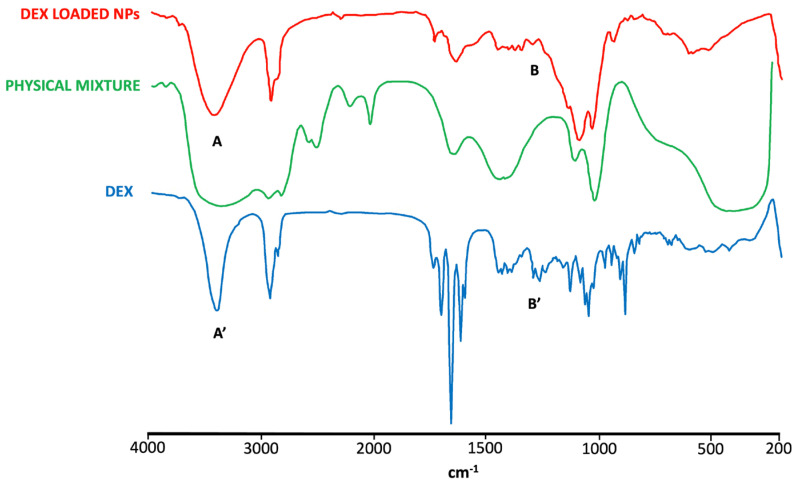
FT-IR spectra of DEX, DEX-loaded NPs, and physical mixture. The characteristic peaks of DEX are altered in the spectrum of DEX-loaded NPs. Specifically, the peak at 3390 cm^−1^ (A) is broadened compared to that shown in the DEX spectrum (A′), indicating the interaction of the drug with CD. Similarly, the peaks in the range of 1268-1400 cm^−1^ (B) are suppressed or absent in the spectrum of DEX-loaded NPs, in contrast to the well-defined peaks present in the DEX spectrum (B′), indicating a strong interaction of the drug with CD.

**Figure 6 pharmaceutics-16-00277-f006:**
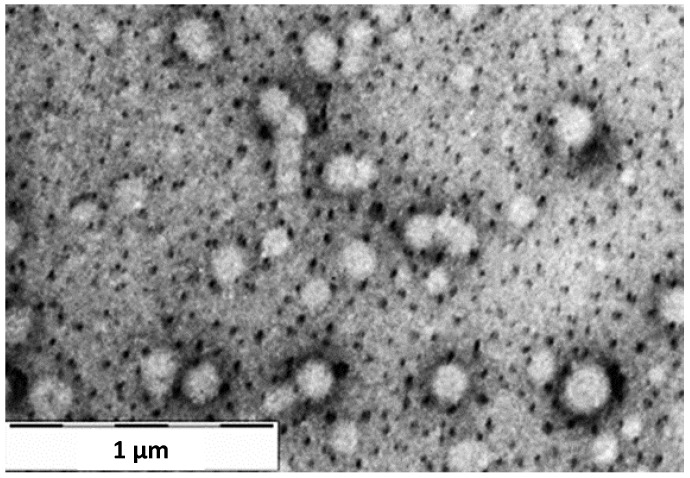
TEM image of DEX-loaded NPs sample at 1 µm magnification.

**Figure 7 pharmaceutics-16-00277-f007:**
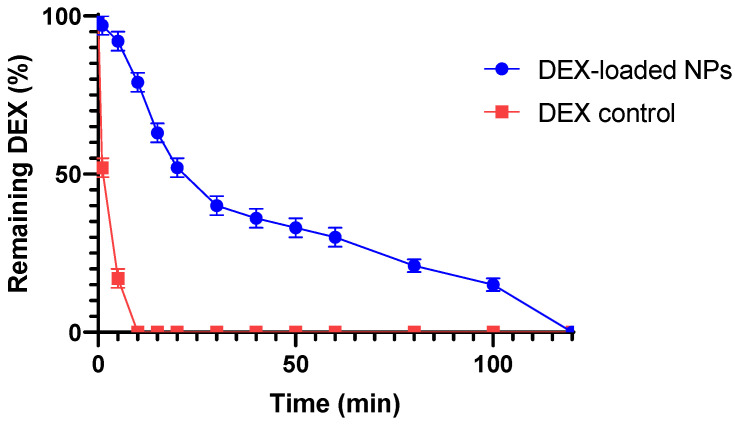
Wash-off mucoadhesion test: slow and gradual decrease in the amount of DEX remaining on the cornea over 120 min shown by the DEX-loaded NPs sample in comparison with the standard control. All values are expressed as the mean of three measurements.

**Figure 8 pharmaceutics-16-00277-f008:**
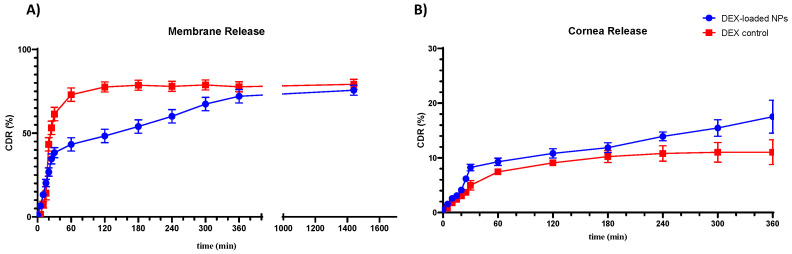
DEX-loaded NPs sample and control solution permeation through Spectra/Por 3 standard regenerated cellulose membrane (MWCO 3.5 kDa) (**A**) and bovine corneal tissue (**B**). All values are expressed as the mean of three measurements.

**Figure 9 pharmaceutics-16-00277-f009:**
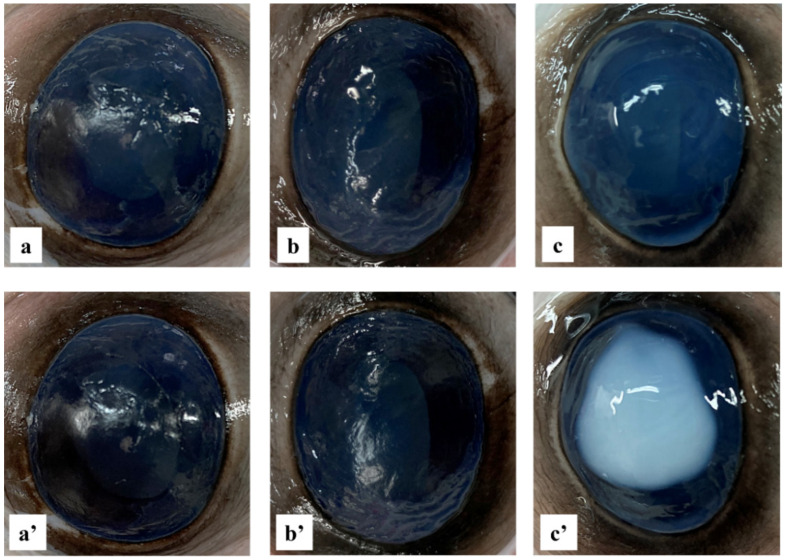
Bovine cornea ocular irritation test. Images of bovine eyes at time zero (**a**–**c**) and after 4 h (**a′**–**c′**), treated with 0.9% NaCl solution (negative control) (**a**,**a′**), DEX-loaded NPs samples solution (**b**,**b′**), and 0.1 N NaOH solution (positive control) (**c**,**c′**).

**Table 1 pharmaceutics-16-00277-t001:** Used volumes of SBE-β-CD and CS solutions in NPs preparation.

Samples	1% *w/v* SBE-β-CD Solution (mL)	0.5% *w/v* CS Solution (mL)
Low Mw CS NPs	1	1.4
Low Mw CS NPs	2	5
Medium Mw CS NPs	1	1.4
Medium Mw CS NPs	2	5
High Mw CS NPs	1	1.4
High Mw CS NPs	2	5
Oligomer CS NPs	1	1.4
Oligomer CS NPs	2	5

**Table 2 pharmaceutics-16-00277-t002:** Equations of kinetic models used for analysis of samples’ release data.

Kinetic Model	Equations	
Zero Order	$D_{t}=D_{0}+K_{0}t$	*D_t_* =amount of drug dissolved in time *t*; *D*_0_ = initial amount of drug in solution; *K*_0_ = zero-order release constant;
First Order	$\log C=\log C_{0}-Kt/2.303$	*C*_0_ = initial drug concentration; *K* = first-order rate constant, *t* = time;
Higuchi	$f_{t=}K_{H}\sqrt{t}$	*K_H_* = Higuchi dissolution constant, *t* = time;
Hixson–Crowell	$\sqrt[{3}]{1-f_{i}}=1-K_{\beta }t$	*f_i_* = fraction of drug dissolved in time *t*; *K_β_* = release constant;
Korsmeyer–Peppas	$\frac{M_{t}}{M_{\infty }}=Kt^{ɳ}$	*M_t_*/*M_∞_* = fraction of drug released at time t; *K* = release rate constant, *ɳ* = release exponent.

**Table 3 pharmaceutics-16-00277-t003:** NPs diameter, PDI, and ζ-Potential values obtained through DLS characterization.

Formulation	Diameter (nm)	PDI	ζ-Potential (mV)	EE (%)
Low MW CS NPs	210.3 ± 5.2	0.161 ± 0.047	+29.4 ± 0.7	/
Medium MW CS NPs	342.7 ± 9.3	0.327 ± 0.121	+17.1 ± 0.5	/
High MW CS NPs	391.1 ± 12.3	0.359 ± 0.174	+19.6 ± 0.8	/
Oligomer CS NPs	/	/	/	/
DEX-loaded Low MW CS NPs	212.9 ± 5.3	0.155 ± 0.048	+31.7 ± 0.4	87.1

**Table 4 pharmaceutics-16-00277-t004:** Obtained values of R^2^ and *n* of DEX-loaded CS/SBE-β-CD NPs samples at different time intervals in corneal release.

Kinetic Model	R^2^	*n*	R^2^	*n*
	0–60 min		60–360 min	
Zero Order	0.9847		0.9967	
First Order	0.9963		0.9653	
Higuchi	0.9377		0.9703	
Hixson–Crowell	0.9938		0.9793	
Korsmeyer–Peppas	0.9988	1.373	0.9612	2.502

**Table 5 pharmaceutics-16-00277-t005:** Ex vivo permeation studies in bovine’s cornea.

Formulation	Flux (J) µg h^−1^ cm^−2^	Apparent Permeability Coefficient P_app_ × 10^−6^ (cm/s)	Transport Enhancement Ratio R = [P_app_(s)/P_app_(c)]
DEX-loaded NPs	6.64	2.13 ± 0.08	2.04
Control	3.22	1.04 ± 0.02	/

**Table 6 pharmaceutics-16-00277-t006:** Bi-directional transport across MDCKII-MDR1 of obtained samples.

Formulation	P_app_ AP-BL × 10^−7^ (cm/s)	P_app_ BL-AP × 10^−7^ (cm/s)	ER (P_app_BL/P_app_AP)
DEX	0.66 ± 0.05	1.87 ± 0.07	2.84
DEX-loaded NPs	0.92 ± 0.03	0.85 ± 0.03	0.92
Diazepam	146 ± 10	123 ± 9	0.84
FD4	10 ± 1	2.0 ± 0.2	0.2

## Data Availability

The data presented in this study are available in this article and Appendix A.

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
