# Peer review of "Chitosan and Anionic Solubility Enhancer Sulfobutylether-β-Cyclodextrin-Based Nanoparticles as Dexamethasone Ophthalmic Delivery System for Anti-Inflammatory Therapy"

_pharmaceutics, 2024, doi:10.3390/pharmaceutics16020277_

Round 1
Reviewer 1 Report
Comments and Suggestions for Authors
Dear authors,
• The authors developed DEX-loaded NPs using the ionotropic gelation of the cationic mucoadhesive chitosan and the anionic solubility enhancer SBE-β-CD. They showed that the solubility of DEX, which is poorly water-soluble, was greatly enhanced by forming an inclusion complex with SBE-β-CD. They characterized the inclusion complex by phase solubility studies and 1H-NMR analysis. They also confirmed that the drug was efficiently complexed and encapsulated within the NPs. They evaluated the chemical and physical properties of the DEX-loaded CS/SBE-β-CD NPs and performed an ex-vivo study on animal mucosa to check the mucoadhesive properties of the final product.
Here are some recommendations:
Table 1: It recommended to specified concentration of SBE-β-CD and CS solutions in NPs preparation.
- Please reference the literature for the” Ex-vivo mucoadhesive studies” methodology. It is recommended that the main form of the setup of this experiment be added next to the schematic image.
- Some reference reports dexamethasone is a BCS class II drug with an aqueous solubility of 0.0505 mg/mL. How did the authors explain this difference?
- It is unclear how the design of Table 1 leads to optimum nanoparticle preparation conditions. It is recommended to explain about this issue.
- It is necessary to add nanoparticle polymers’ FTIR graphs. Also, the quality of the figure should be increased and the defining peaks on the shape will be determined.
- For describing crystallinity and interaction between compounds, DSC of all components in the nanoparticle should be added.
- It seemed “post cataract surgery” is mostly unnecessary in the title. It is recommended to revise the title so that illustrates “anionic solubility enhancer SBE-β-CD “.
- It recommended authors add “ formation of the inclusion complex and its stoichiometry were studied through the phase solubility studies, Job's Plot method, and Bi-directional transport studies on MDCKII-MDR1” in the abstract as authors mentioned these tests as the strength.
Regards,
Author Response
The authors thank the reviewer for her/his comments. Please refer to the attached file for the responses.

Reviewer 2 Report
Comments and Suggestions for Authors
see attached file

Author Response

(The authors gave the same response as above.)

Reviewer 3 Report
Comments and Suggestions for Authors
In section 2.1 the Authors presented that oligomer chitosan has molecular weight 13.49 kDa, whereas in section 2.5 oligomer CS has 2 kDa. Which value is correct?
The nanoparticles have a diameter up to 100 nm. The DLS analysis demonstrates the diameter over 200 nm for the particles, so they cannot be called nanoparticles. What is the diameter in TEM microphotographs?
The Authors should also present the Polysorbate 80 FTIR spectrum? Is it a component of nanoparticles?
If the characteristic peaks for pure DEX are not wisible in DEX-LOADED NPS DLS and FTIR analyses, how the Authors can confirm the presence of the drug in the complex?
Author Response

(The authors gave the same response as above.)

Reviewer 4 Report
Comments and Suggestions for Authors
It is a rare case, when I do not wish to improve something.
Good luck.
Author Response
The authors are very happy with the comment and sincerely thank the reviewer.
Round 2
Reviewer 3 Report
Comments and Suggestions for Authors
Thank you for the answers. The article should be published.